# Thermography as a Non-Ionizing Quantitative Tool for Diagnosing Burning Mouth Syndrome: Case-Control Study

**DOI:** 10.3390/ijerph19158903

**Published:** 2022-07-22

**Authors:** Elena Nicolas-Rodriguez, Ana Garcia-Martinez, Diana Molino-Pagan, Luis Marin-Martinez, Eduardo Pons-Fuster, Pia López-Jornet

**Affiliations:** 1Faculty of Medicine and Odontology, Hospital Morales Meseguer, Clínica Odontológica, Marqués de los Vélez s/n, 30008 Murcia, Spain; elena.nicolasr@um.es (E.N.-R.); ana.garcia27@um.es (A.G.-M.); 2Collaborate Faculty of Medicine and Odontology, Hospital Morales Meseguer, Clínica Odontológica, Marqués del los Vélez s/n, 30008 Murcia, Spain; dyana_mp@hotmail.com; 3Servicio de Endocrinología y Nutrición, Hospital General Universitario Santa Lucía, Urb. Novo Carthago, 80, 30202 Cartagena, Spain; luis.marin.martinez@gmail.com; 4Departamento de Anatomía Humana y Psicobiología, Faculty of Medicine and Odontology, Biomedical Research Institute (IMIB-Arrixaca), University of Murcia Spain, 30100 Murcia, Spain; eduardo.p.f@um.es; 5Faculty of Medicine and Odontology, Biomedical Research Institute (IMIB-Arrixaca), Hospital Morales Meseguer, Clínica Odontológica, Marqués del los Vélez s/n, 30008 Murcia, Spain

**Keywords:** burning mouth syndrome, infrared thermography, tongue temperature

## Abstract

Objectives: Thermography is an imaging technique based on the acquisition and analysis of thermal data. The present study evaluates the use of tongue infrared thermography (IRT) as a tool for the diagnosis of burning mouth syndrome (BMS). Material and methods: An IRT study was carried out in patients diagnosed with BMS according to the criteria of the International Association for the Study of Pain (*n* = 32) and in healthy controls (*n* = 35). Burning sensations, dry mouth and taste disturbances were assessed, and three temperature values were recorded for each tongue surface (dorsal, right lateral, left lateral and tip), along with body temperature and environmental temperature. Results: A statistically significant difference was recorded in the temperature of the dorsal surface of the tongue between the BMS group and the controls (*p* = 0.01). The area under the curve (AUC) was 0.731 (95% CI: 0.402–0.657; *p* = 0.003). The sensitivity and specificity obtained was 62% and 77%, respectively. Conclusions: Infrared thermography appears to be useful as a complementary tool for the diagnosis of BMS, though further studies are needed in this field.

## 1. Introduction

Burning mouth syndrome (BMS), also known as oral dysesthesia, glossodynia or glossopyrosis [1,2,3], is a chronic disorder of the oral cavity defined by the International Association for the Study of Pain as a burning sensation of the oral mucosa, in which the symptoms manifest for more than two hours a day for a period of over three months, in the absence of apparent clinical lesions [1,2,3,4,5]. There are different classifications of BMS. Scala et al. [2] distinguished between primary (or idiopathic) BMS, when the underlying cause is not known, and secondary BMS, when the disorder is associated with systemic diseases and/or local factors. The etiology of burning mouth syndrome is complex and multifactorial, and has not been fully clarified to date [4,5,6,7,8,9,10,11,12,13,14,15,16,17,18].

The burning or itching pain of BMS usually manifests in any region of the oral mucosa, and may affect either a single location or a number of different mucosal zones at the same time [7]. The discomfort typically increases over the course of the day, reaching maximum intensity in the afternoon or evening, and disappears during sleep [11,12]. Affected patients experience worsening of the symptoms with hot beverages, while cold beverages and eating afford symptom relief [12]. The pain may be associated with other subjective symptoms such as dry mouth (xerostomia) and taste disturbances [1,1]. In this regard, Just et al. described a decrease in taste function in patients with BMS [15], while Imura et al. recorded no significant changes in the taste identification thresholds [19].

Infrared thermography (IRT) is a technique that allows measurement of the infrared radiation emitted by an object at a distance, and displays the temperature of the object onscreen [20,21]. The rationale for the use of IRT in biomedicine is that changes in body temperature may be indicative of the existence of disease conditions. Such temperature changes are related to the inflammation produced by the disease, and can be detected and recorded with a thermographic camera [20,21,22,23,24,25,26,27,28,29,30,31,32,33]. Thermal (infrared) imaging offers a number of advantages: it is noninvasive, involves no physical contact with the measured surface, and affords images that can be filed and processed to evaluate neurological disorders (e.g., spinal cord and peripheral nerve damage), vascular diseases (e.g., Raynaud’s disease) and bone and joint disorders (e.g., myofascial pain syndrome, rheumatoid arthritis and sports injuries). In addition, it can be used to establish an early diagnosis of breast cancer, to assess infertility, analyze thyroid gland nodules, and diagnose periapical dental infections [33]. Likewise, the technique can be employed to monitor changes in bone temperature during drilling for preparation of the dental implant bed, etc.

Thermography is an imaging technique with a promising future that offers new information about patients in many areas referred to in the head and neck region. As an example, it has been used in patients with Bell’s palsy, where asymmetrical temperature values have been documented on both sides of the face, with a sensitivity of 0.867 and a specificity of 0.800. In this regard, IRT has been reported to be useful in establishing an early diagnosis of this type of palsy [34].

Thermography has also been used for the diagnosis of temporomandibular joint (TMJ) disorders. In this regard, patients with asymptomatic TMJ disorders have been shown to present symmetrical thermal patterns with a mean standard deviation of 0.1, while individuals with TMJ pain present asymmetrical patterns with increased temperature on the affected joint, with a mean standard deviation of 0.4 [35,36]. However, Machoy et al. [37], in a review of the diagnosis of TMJ disorders, indicated that there is no standardized protocol for measuring the temperature of the masticatory muscles based on infrared imaging. This makes it difficult to use the technique as a standard clinical diagnostic tool capable of drawing solid conclusions from the images obtained.

The tongue is a key organ capable of reflecting disease conditions in the body. Indeed, the clinical evaluation of the tongue is used as part of the diagnostic process in different diseases, based on its color, shape, humidity and thickness [27,28,29,30]. However, at present, temperature assessment using IRT is being introduced, fundamentally for the diagnosis of diabetes [28] and anemia, based on local temperature modifications secondary to changes in blood flow [29].

Burning mouth syndrome remains a challenge for clinicians, since there is no gold standard for establishing its diagnosis. In effect, the diagnosis of BMS is based on the exclusion of other possible explanations for the patient manifestations. In this regard, the present study was carried out to evaluate lingual infrared thermography as a noninvasive tool for the diagnosis of BMS.

## 2. Materials and Methods

The study was approved by the Research Ethics Committee of the University of Murcia (Murcia, Spain ID: 3608/2021). Written informed consent was obtained from all the participants in the study, in abidance with the Declaration of Helsinki. A cross-sectional case-control design was used, in which infrared thermography was performed in patients with BMS and in healthy controls. The study was carried out in line with the STROBE (Strengthening the Reporting of Observational Studies in Epidemiology) statement.

### 2.1. Participants

We included 32 consecutive patients from the Dental Clinic of the University of Murcia (Morales Meseguer Hospital), diagnosed with BMS according to the criteria of the International Classification of Orofacial Pain [3].

Individuals under 18 years were excluded, as were those with oral disease, dental alterations or geographic tongue, depapillation, fever, infectious processes, decompensated systemic disease, neck or face traumatisms, and pregnant women.

The control group in turn consisted of healthy individuals seen in the Dental Clinic, without oral or dental disease, and with the same age and gender distribution as in the group of patients with BMS.

A single trained investigator used a customized questionnaire to record demographic data (age and gender), disease history, and toxic habits (e.g., smoking: yes/no). The following symptoms were registered:Burning sensation, using a visual analogue scale (VAS) from 0–10 (0 = no burning sensation, 10 = extreme burning sensation).Dry mouth or xerostomia, using a VAS from 0–10 (0 = no dry mouth sensation, 10 = extreme dry mouth sensation).Taste alterations, defined as perceived changes in taste quality (metallic, bitter, sweet, salty and acid).

### 2.2. Recording of Thermographic Images

The IRT recordings were carried out in a room with constant environmental temperature, with the patients under resting conditions and without having performed physical exercise in the last hour. Likewise, the patients were required to have consumed no hot or cold beverages or food during the hour before the recordings. With the patients in the sitting position, the examiner maintained a distance of 15–20 cm between the target body surface (tongue) and the thermographic device. The temperatures of each zone were recorded in the absence of any physical contact with the patient, in order to avoid heat transfer between bodies and distortion of the measurements. Three temperature values were recorded for each tongue region (dorsal, right lateral, left lateral and tip), and body temperature and environmental temperature were also recorded. A thermographic image was obtained for each tongue region (Figure 1). 

The tongue temperature recordings were made using a manual thermographic camera (model HT−02D, Hti Dongguan Xintai Instrument Co., Ltd., Donggaun, China), with a measurement precision of ±2% and a thermal sensitivity of 0.3 °C. The operating wavelength was between 8–11.5 µm, with a field of view of 33° × 33°, and a minimum focal distance of 0.5 m. Emissivity was adjustable between 0.1–1.0, and the image resolution was 32 × 32 (1024 pixels).

Emissivity is the ratio between the radiation emitted by the surface of a body and the radiation emitted by a black body at the same temperature. In the case of the human body, emissivity is considered to range between 0.94–0.99. In the present study the value was established as 0.95. A rainbow color palette was used to obtain the images, representing a color scale from blue to red.

### 2.3. Statistical Analysis

Descriptive statistics were calculated for quantitative (mean and standard deviation [SD]) and qualitative variables (frequencies and percentages). The Pearson correlation test was used to explore associations between the different study variables. The Student t-test was applied to compare results according to study group, and analysis of variance (ANOVA) was used to compare results according to taste alteration. Effect size was estimated for each variable by following the suggestions of Dominguez-Lara [33], according to the homogeneity characteristics of the sample. Receiver operating characteristic (ROC) curves were plotted to determine sensitivity and specificity. Statistical significance was considered as *p* ≤ 0.05. The SPSS version 25.0 statistical package (IBM, Armonk, NY, USA) was used throughout.

## 3. Results

The study sample consisted of 67 individuals (35 controls and 32 patients with BMS), of which 83.6% were women and 16.4% men. The BMS group (29 women and 3 men) had a mean age of 65.09 ± 11.0 years, while the control group (27 women and 8 men) had a mean age of 55.23 ± 9.9 years. Five of the participants were active smokers (1 patient with BMS and 4 controls).

A statistically significant difference was recorded in the temperature of the dorsal surface (*p* = 0.01) and right lateral surface of the tongue (*p* = 0.029) between the BMS group and the controls (Table 1).

The mean burning sensation VAS score in the BMS group was 7.03 ± 2.0, while the mean dry mouth or xerostomia score was 6.69 ± 3.0. Twenty-five percent of the patients reported no taste alterations. Among the patients who did describe taste alterations, a metallic taste was seen to predominate (34.4% of the cases), followed by a bitter taste (15.6%). On the other hand, 18.8% of the patients reported dysgeusia, while 6.3% claimed to perceive various tastes.

The different measured surfaces showed an association between burning sensation and body temperature (r = 0.54; *p* ≤ 0.05) and between the temperature of the tip of the tongue and dry mouth sensation (r = 0.47; *p* ≤ 0.05). Likewise, a statistically significant association was found between taste alteration and the temperature of the dorsal surface of the tongue (*p* = 0.038) (Table 2).

The area under the curve (AUC) was 0.731 (95% confidence interval (95% CI]: 0.402–0.657; *p* = 0.003) (Figure 2). The temperature with a cut-off point of 33.7 °C showed a sensitivity and specificity of 62% and 77%, respectively, with a likelihood ratio of 0.532.

## 4. Discussion

In the present study, higher temperatures were recorded in the patients with BMS versus the controls, and significant differences were found in the case of the temperature of the dorsal surface of the tongue. The advantages of thermography are the noninvasive nature of the technique, asepsis, the absence of ionizing radiation, and the relatively low cost of the procedure [20,21,22]. In this regard, the present study raises interesting options for the use of IRT as an innocuous tool for the diagnosis on BMS.

Ammoush et al. [38] evaluated the thermal imaging data obtained in facial cellulite and dental abscesses, and recorded significant temperature differences in both disorders. Specifically, the temperature difference between the affected side and the non-affected side was 1.49  ±  1.0 in the case of dental abscesses and 2.4  ±  1.9 in the case of facial cellulite. The greater difference in patients with cellulite versus those with dental abscesses was attributed to the comparatively greater spread and tissue damage produced by cellulite.

Thermography has also been investigated in periapical inflammatory lesions. Aboushady et al. [39] studied 80 patients to assess the validity of the technique versus the reference standard, with application to acute pulpitis with apical periodontitis, acute periapical abscess, and chronic periapical abscess—the highest temperature recordings were found to correspond to acute periapical abscess. Thus, the technique could be useful for establishing an early diagnosis of inflammatory disorders, even before the appearance of clinical symptoms.

On the other hand, the use of IRT has been investigated in inflammatory processes such as prosthetic stomatitis associated with Candida infection. In a study of individuals subjected to clinical examination, microbiological culture and thermographic analysis of the palatal mucosa, the data showed the healthy controls to have significantly lower temperature values than the patients with stomatitis [40].

Altered body temperature is a natural indicator of disease [20,21,22], and in this respect tongue temperature is a key oral physiological parameter that has been shown to affect food taste and texture sensation. Thus, IRT may be useful for exploring patients’ sensory perception [31]. In our study, we recorded a statistically significant association between taste disturbances in patients with BMS and the temperature of the dorsal surface of the tongue (*p* = 0.038).

The measurement of tongue temperature is clinically useful since the tongue has an important blood supply, and its surface temperature reflects the internal (core) temperature transferred through the bloodstream. Hence, a decrease in tongue temperature could be indicative of a decrease in the blood perfusion indices [41,42]. In concordance with the observations of Jiang et al. [41], the temperature of the dorsal surface of the tongue in our patients was higher than that of the lateral surfaces, which in turn proved similar on both sides (right lateral and left lateral surfaces of the tongue), followed by the tip of the tongue, where the temperature was lower than in the rest of the tongue regions.

The etiopathogenesis of BMS remains unclear, though a number of hypotheses have been proposed to explain the origin of the pain in patients with this syndrome. Wolowski et al. [43] used a standardized battery of quantitative sensory tests in patients with BMS, and reported that the syndrome could be a combination of dysfunction of the free nociceptive nerve endings in the peripheral nervous system and altered pain processing at the central level. Madariaga et al. [12], in a meta-analysis, reported that thermal sensitivity seems to be altered in patients with BMS compared to the controls, suggesting the existence of small-fiber neuropathy, with significant differences between the patients with BMS and the controls in terms of the heat (effect size = 0.683; *p* < 0.05) and cold detection thresholds (effect size = −0.580; *p* < 0.001).

In relation to the temperature of the dorsal surface of the tongue, our study showed IRT to have a sensitivity of 62.5% and a specificity of 77.1%. The advantages of thermography are the noninvasive nature of the technique, asepsis, the absence of ionizing radiation, and the relatively low cost of the procedure. In addition, IRT is a contactless procedure, which is very useful for infection control, and allows us to compare different regions of interest in two-dimensional images in real time [20,21,22,43].

The diagnosis of BMS is based on exclusion, and in this respect, we need to discard other possible oral lesions, systemic disorders or infections that could give rise to similar manifestations [1,2,3,4,5]. In this setting, the presence of thermal changes could serve as a guide for evaluating BMS. In any case, it must be taken into account that IRT cannot be used as an independent or absolute test for diagnosing BMS, and that the results obtained with the technique must be interpreted within the clinical context of the patient. Provided adequate standards are applied, IRT may offer crucial information that complements the data obtained with the existing methods. 

With regard to the limitations of our study, it must be taken into account that a number of technical difficulties remain that complicate the recording of tongue temperature. On the other hand, this is a single-center observational study, and future prospective trials conducted in the routine clinical practice setting are needed to analyze the changes over time, with a view to better understanding this syndrome.

In conclusion, IRT offers important advantages, including the rapidity of the technique and its noninvasive and non-ionizing nature. 

## 5. Conclusions

Thermography of the dorsal surface of the tongue may serve as a complementary tool in the diagnosis of BMS, though standardized protocols are needed, as well as further studies, in order to confirm the potential usefulness of the technique.

## Figures and Tables

**Figure 1 ijerph-19-08903-f001:**
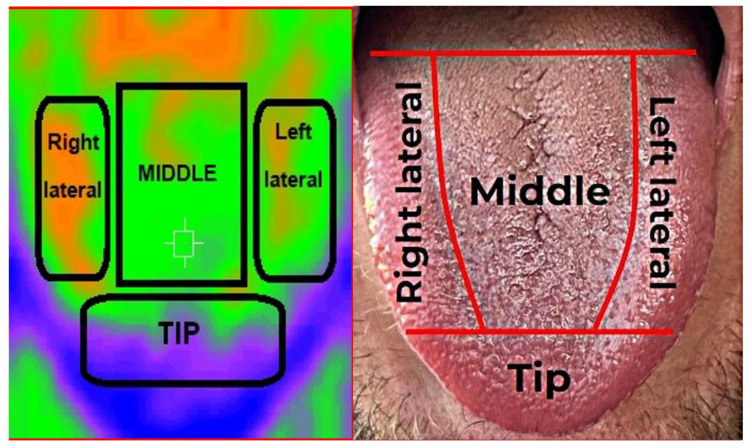
Illustration of the areas of the tongue surface and infrared thermal images.

**Figure 2 ijerph-19-08903-f002:**
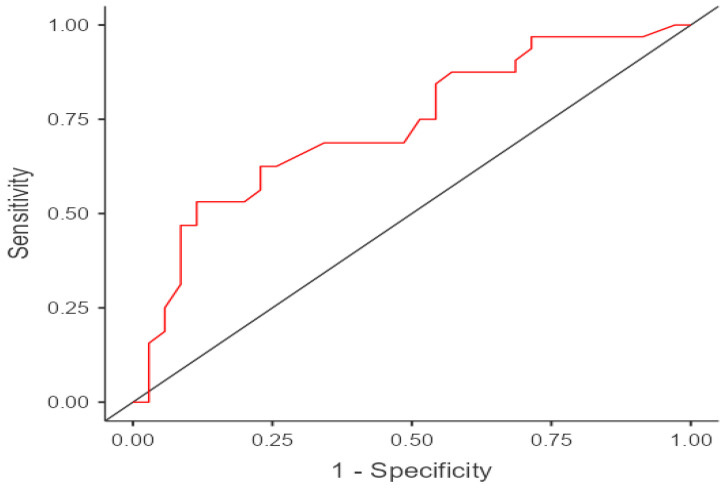
Curve ROC The area under the curve (AUC) was 0.731 (95% CI: 0.402–0.657; *p* = 0.003). The sensitivity and specificity obtained was 62% and 77%, respectively.

**Table 1 ijerph-19-08903-t001:** Temperature distribution by study group (control and burning mouth syndrome).

Variables Tongue	Control (*n* = 35)	Burning Mouth (*n* = 32)	*p* Value
	Mean ± ST	Mean ± ST	
TªM Dorsum	33.33 ± 1.0	34.12 ± 0.9	0.001 *
TªM Lateral right	32.81 ± 1.0	33.32 ± 0.8	0.029 *
TªM Lateral left	32.73 ± 1.0	33.07 ± 0.9	0.146
TªM apex	31.27 ± 1.2	31.43 ± 1.2	0.600
body temperature	33.71 ± 0.5	33.83 ± 0.4	0.288

Note: TªM: Temperature ST standard deviation; * *p* < 0.05.

**Table 2 ijerph-19-08903-t002:** Taste alteration distribution in burning mouth syndrome patients.

Variable	Taste Alteration					*p* Value
	No Alterations (*n* = 8) Mean ± ST	Metallic (*n* = 11) Mean ± ST	Bitter (*n* = 5) Mean ± ST	Disgeusia (*n* = 6) Mean ± ST	Others (*n* = 2) Mean ± ST	
TªM Dorsum	34.39 ± 0.8	34.16 ± 0.8	34.15 ± 0.6	33.20 ± 0.8	35.10	0.038 *
TªM Lateral right	33.56 ± 0.7	33.28 ± 0.8	33.33 ± 0.9	32.73 ± 0.9	34.32 ± 0.5	0.196
TªM Lateral left	33.29 ± 0.8	33.04 ± 0.9	33.19 ± 0.6	32.29 ± 0.8	34.35 ± 0.2	0.072
TªM Apex	31.45 ± 1.4	31.27 ± 0.8	32.33 ± 1.1	30.59 ± 1.3	32.75 ± 1.5	0.171
TªM body	33.90 ± 0.4	33.94 ± 0.3	33.59 ± 0.4	33.69 ± 0.4	33.98	0.181
TªM environmental	21.28 ± 2.1	21.63 ± 1.1	21.35 ± 1.3	22.60 ± 1.0	22.75 ± 0.8	0.203

TªM: Temperature ST standard deviation; * *p* < 0.05.

## Data Availability

The datasets generated during and/or analyzed during the current study are available from the corresponding author on reasonable request.

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
