# Peer review of "Thermography as a Non-Ionizing Quantitative Tool for Diagnosing Burning Mouth Syndrome: Case-Control Study"

_ijerph, 2022, doi:10.3390/ijerph19158903_

Round 1

Reviewer 1 Report

Dear Author

Excellent article, well written.

I would recommend that you frame the examples of applications of thermographic imaging in healthcare included in the discussion in a way that is more relevant to the work. The paper reads very well except for this section as the examples are presented with little context or emphasis on the relationship with the research presented.

Lines 176-208 present research related to the general field but it feels like this kind of presentation would be better suited to an introduction. In the discussion, each of these pieces of research should be mentioned when they are relevant to the work presented here and at that point in the discussion where it is relevant to do so.

Kind regards

Author Response

Thank you for your helpful comments. . In accordance with your suggestion, we have revised the text 

Lines 176-208 present research related to the general field, but it seems that this type of presentation would be more appropriate for an introduction.  I have been added to introduction 

We have emphasized the aspects of the discussion. We thank you for this suggestion

Reviewer 2 Report

This work evaluates the use of tongue infrared thermography as a tool for the diagnosis burning mouth syndrome(BMS). The experiemental results demonstrated that infrard thermography appears to be useful as a complementary tool for the diagnosis of BMS. This work is very interesting, and the timeless and novelty is quality for publishing in international journal of Environmental research and public health. However, there are seveal problem need to be addressed,

(1) It is suggested to supplement the actual detection results of infrared imaging.

(2) The English needs to be polished.

(3) Whether feature extraction algorithm is needed to improve detection SNR?

Author Response

Thank you for your helpful comments.  In accordance with your suggestion, we have revised infrared imaging.

Yes, standardized protocols are needed to improve imaging results in BMS

Reviewer 3 Report

The only comment thins reviewer has is regarding figure1. For the benefit of the reader the authors are requested to demarcate different regions of the tongue on the thermal map as well.  

Author Response

We are grateful to you and the reviewers for providing valuable comments, which have been instrumental in improving our manuscript

As per your suggestion, we have reviewed the infrared image.

Round 2

Reviewer 2 Report

None